# Identification of the Neurokinin-1 Receptor as Targetable Stratification Factor for Drug Repurposing in Pancreatic Cancer

**DOI:** 10.3390/cancers13112703

**Published:** 2021-05-30

**Authors:** Iris Beirith, Bernhard W. Renz, Shristee Mudusetti, Natalja Sergejewna Ring, Julian Kolorz, Dominik Koch, Alexandr V. Bazhin, Michael Berger, Jing Wang, Martin K. Angele, Jan G. D’Haese, Markus O. Guba, Hanno Niess, Joachim Andrassy, Jens Werner, Matthias Ilmer

**Affiliations:** 1Department of General, Visceral and Transplantation Surgery, University Hospital, Ludwig-Maximilians-University Munich, 81377 Munich, Germany; iris.beirith@med.uni-muenchen.de (I.B.); bernhard.renz@med.uni-muenchen.de (B.W.R.); shristee.mudusetti@med.uni-muenchen.de (S.M.); natalja.ring@med.uni-muenchen.de (N.S.R.); dominik.koch@med.uni-muenchen.de (D.K.); alexandr.bazhin@med.uni-muenchen.de (A.V.B.); drjwang@ustc.edu.cn (J.W.); martin.angele@med.uni-muenchen.de (M.K.A.); jan.dhaese@med.uni-muenchen.de (J.G.D.); markus.guba@med.uni-muenchen.de (M.O.G.); Hanno.niess@med.uni-muenchen.de (H.N.); joachim.andrassy@med.uni-muenchen.de (J.A.); jens.werner@med.uni-muenchen.de (J.W.); 2German Center for Translations Cancer Research (DKTK), Partner Site Munich, 80336 Munich, Germany; 3Department of Pediatric Surgery, Research Laboratories, von Hauner Children’s Hospital, Ludwig-Maximilians-University Munich, 80337 Munich, Germany; julian.kolorz@med.uni-muenchen.de (J.K.); michael.berger@med.uni-muenchen.de (M.B.); 4Department of General, Abdominal and Transplant Surgery, Essen University Hospital, 45417 Essen, Germany; 5Department of General Surgery, The First Affiliated Hospital of USTC, Division of Life Sciences and Medicine, University of Science and Technology of China, Hefei 230036, China

**Keywords:** pancreatic ductal adenocarcinoma, PDAC, neurokinin-1 receptor, NK1R, TACR1, substance P, aprepitant, SP/NK1R-complex

## Abstract

**Simple Summary:**

Pancreatic ductal adenocarcinoma is the most common form of pancreatic cancer. It is known for low life expectancies after diagnosis and very limited treatment options. The identification of new therapeutic molecular targets is urgent as it might allow faster development of new treatment strategies. Targeting the neurokinin-1 receptor with small molecules has previously shown anti-tumoral effects in a large variety of cancers. Here, we found specific types of pancreatic cells to express the neurokinin-1 receptor while at the same time showing positive treatment response represented by cell growth reduction in number and size when treated with aprepitant. Our results suggest the neurokinin-1 receptor as a promising targetable structure and therefore interesting in the concept of personalized medicine.

**Abstract:**

The SP/NK1R-complex plays an important role in tumor proliferation. Targeting of the neurokinin-1 receptor in previous studies with its antagonist aprepitant (AP) resulted in anti-tumoral effects in colorectal cancer and hepatoblastoma. However, there is still a lack of knowledge regarding its effects on pancreatic cancer. Therefore, we treated human pancreatic ductal adenocarcinoma (PDAC) cell lines (Capan-1, DanG, HuP-T3, Panc-1, and MIA PaCa-2) and their cancer stem cell-like cells (CSCs) with AP and analyzed functional effects by MTT-, colony, and sphere formation assays, respectively; moreover, we monitored downstream mechanisms by flow cytometry. NK1R inhibition resulted in dose-dependent growth reduction in both CSCs and non-CSCs without induction of apoptosis in most PDAC cell lines. More importantly, we identified striking AP dependent cell cycle arrest in all parental cells. Furthermore, gene expression and the importance of key genes in PDAC tumorigenesis were analyzed combining RT-qPCR in eight PDAC cell lines with publicly available datasets (TCGA, GEO, CCLE). Surprisingly, we found a better overall survival in patients with high NK1R levels, while at the same time, NK1R was significantly decreased in PDAC tissue compared to normal tissue. Interestingly, there is currently no differentiation between the isoforms of NK1R (truncated and full; NK1R-tr and -fl) in any of the indicated public transcriptomic records, although many publications already emphasize on important regulatory differences between the two isoforms of NK1R in many cancer entities. In conclusion, analysis of splice variants might potentially lead to a stratification of PDAC patients for NK1R-directed therapies. Furthermore, we presume PDAC patients with high expressions of NK1R-tr might benefit from treatment with AP to improve chemoresistance. Therefore, analysis of splice variants might potentially lead to a stratification of PDAC patients for NK1R-directed therapies.

## 1. Introduction

Pancreatic ductal adenocarcinoma (PDAC) is known for its frequent late diagnosis at advanced stages of cancer progression. With a 5-year survival rate of less than 10%, it is the most aggressive form of pancreatic cancer [1,2]. The incidences are expected to rise about 3% per year, causing scientists to expect PDAC to be the second leading cause of cancer-related death by 2030 [3,4]. 

The pancreas is characterized by highly complex interactions between the nervous system and disease [5]. With PDAC being innervated by both sympathetic, parasympathetic, and sensory nerves, its hallmarks comprise increased neural density as well as neural hypertrophy [6,7,8]. Currently, the recruitment of nerves is an emerging hallmark of cancer, and multiple pharmacological approaches are investigated to influence their signaling in the tumor microenvironment serving as a promising novel therapeutic strategy in the treatment of cancer [9].

The targeted inhibition of the substance P/neurokinin 1 receptor (SP/NK1R) system with its critical role in neuroinflammation has been considered as a promising drug target within the scope of personalized medicine [10,11]. SP, released by primary sensory nerve fibers, is a member of the tachykinin family [11,12]. These intensively studied and structurally related neuropeptides are known for being expressed throughout the nervous and immune system and are involved in a myriad of biological and physiological processes, including inflammation and proliferation [10,12,13]. Furthermore, they contribute to multiple pathological conditions, including acute and chronic inflammation, infection, and cancer, among others [12]. 

SP is encoded by *TAC1* and binds to three tachykinin receptors of which NK1R shows the highest binding affinity [10]. NK1R, also referred to as tachykinin receptor 1 (*TACR1*), is a G-protein coupled (GCPR), seven-transmembrane domain receptor [14]. The human receptor exists in two distinct isoforms, evoking diverse functionalities and differential expression across the body. As such, the full-length version consists of 407 aa (NK1R-fl) and is to be found at certain sites in the human brain [15]. SP mediated activation of NK1R-fl results in the assembly of a scaffolding complex incorporating β-arrestin, ERK1/2, and p38MAPK among others promoting proliferative and anti-apoptotic effects [16]. In contrast, the truncated isoform (311 aa; NK1R-tr) lacks at the C-terminus (exon 5), and is predominantly expressed in the central nervous system, as well as in peripheral tissues [15,17]. Receptor truncation leads to a decrease of SP binding affinity by at least 10-fold and inhibits β-arrestin-involved complex formation through failure of NK1R endocytosis [18]. 

The NK1R antagonist aprepitant (AP) has been approved by the FDA for the treatment of chemotherapy-induced nausea and vomiting in low dosage. Latest publications suggest higher dosages of AP to act as anti-tumoral agent through the inhibition of proliferation and induction of apoptosis in a variety of malignant cells [19,20]. We have previously shown that targeting of the SP/NK1R signaling cascade with AP successfully inhibits canonical Wnt signaling, while at the same time causing significant growth reduction in human colon cancer and hepatoblastoma cells [21,22]. However, the mechanisms of AP treatment in pancreatic tumorigenesis are poorly explored. Therefore, we aimed to investigate the effects of treatment with AP and SP on cancer stem-like cells (CSCs) and multiple heterogenic pancreatic cancer cell lines regarding cell proliferation and intracellular mechanisms. 

## 2. Materials and Methods

### 2.1. Cell Culture

We used the following pancreatic cancer cell lines: BxPC-3, Capan-1, DanG, HuP-T3, Panc-1, MIA PaCa-2, PSN-1, and AsPC-1. All cell lines were cultured in the appropriate media according to the ATCC recommendations (Gibco^®^ RPMI 1640 for BxPC-3, Capan-1, DanG, PSN-1, and AsPC-1 or Gibco^®^ Dulbecco’s Modified Eagle’s Medium (DMEM) for the remaining), supplemented with 10% fetal bovine serum (FBS; Corning, Wiesbaden, Germany), and 1% streptomycin/penicillin (PAN-Biotech, Aidenbach, Germany) at 37 °C in a humidified incubator with 5% CO_2_. For all experiments, cell lines were used up to passage number 20. Cell culture medium for pancreatic stellate cells (PSCs) consisted of Gibco^®^ DMEM/F-12, 1% amphotericin B (PAN-Biotech, Aidenbach, Germany), 10% FBS, and 1% streptomycin/penicillin. Detection of mycoplasma was conducted regularly using conventional PCR technique. All cells tested negative for mycoplasma contamination. All cells were authenticated commercially by IDEX BioResearch (Ludwigsburg, Germany). 

### 2.2. Preparation of PSC Conditioned Media

PSC conditioned media (CM) were obtained by culturing primary PSCs for 24 h at approximately 70% confluency. CM were collected and filtered before through a 0.4 μM filter. PDAC cell lines were cultured in a 1:1 ratio of appropriate cell line media and CM for the indicated time. 

### 2.3. Drugs

Aprepitant (Tocris Bioscience, Bristol, UK) (NK1R antagonist) was dissolved at 50 mM in DMSO. Substance P (Tocris Bioscience, Bristol, UK) (NK1R agonist) was dissolved at 1 mM in distilled water. Drugs were stored at −20 °C. 

### 2.4. Viability Assay

Cell viability was assessed using a 3-(4,5-Dimethylthiazol-2-yl)-2,5-Diphenyltertazolium Bromide (MTT) (Invitrogen, Carlsbad, CA, USA) assay, and 15,000 cells were seeded into 96-well plates (NUNC, Langenselbold, Germany). After 24 h, cells were treated with increasing doses of AP (5–50 µM) for 24 h. To assess cell viability, 50 μL of MTT lysis solution with a final concentration of 0.5 mg/mL in 1X PBS (AppliChem, Darmstadt, Germany) was first added to each well followed by 30 min incubation at 37 °C. The MTT solution was discarded and 50 μL of DMSO (Sigma-Aldrich, Taufkirchen, Germany) was added to each well. For the readout, a multi-scanner micro-plate reader (VersaMax Microplate Reader, Molecular Diagnostics, CA, USA) was used to measure the absorbance at 570 nm with a background absorbance of 670 nm. 

### 2.5. Sphere and Colony Formation Culture

Sphere culture and assays were performed as previously described [23]. Briefly, sphere formation medium was prepared with Advanced DMEM/F-12, 1% penicillin/streptomycin, and 1% methylcellulose (Sigma-Aldrich, Taufkirchen, Germany), further supplemented by 10 ng/mL human recombinant βFGF, 20 ng/mL human recombinant EGF, and 1× B27 serum-free supplement (supplies obtained from Invitrogen, Carlsbad, CA, USA). PDAC cells were trypsinized and washed twice with DPBS. An amount of 500 cells were seeded into 96-well ultra-low attachment plates (Corning, Wiesbaden, Germany) in 100 μL sphere formation medium. Before counting, spheres were cultivated for 10–14 days in media containing additives as mentioned in the text (20 μM AP, 100 ng/mL SP or DMSO). Medium was replenished twice per week. 

For colony formation, 500 c/w cells were seeded onto 6-well plates. After incubation for 24 h, treatments with different concentrations of aprepitant and SP (1 µM, 10 µM, 20 µM, 40 µM plus combinations with 20 nM of SP) were started. Colonies were counted after 12 days by staining with crystal violet (CV) (0.1% CV in 20% Methanol) for 20 min, dried overnight and measured at 570/670 nm with Versa Max microplate reader (Molecular Diagnostics, CA, USA). 

### 2.6. RNA Isolation and RT-qPCR

Isolation of RNA was performed using RNeasy Mini Kit (Qiagen, Hilden, Germany). This was followed by cDNA synthesis, realized through QuantiTect Reverse Transcription Kit (Qiagen, Hilden, Germany) using 1 µg of the respective isolated RNA. Thermal cycling during cDNA synthesis was realized by Eppendorf Mastercycler gradient.

PCR reaction was set up employing QuantiNova SYBR Green PCR Kit (Qiagen, Hilden, Germany). qPCR thermal cycling was done through StepOne Real-Time PCR System (Applied Biosystems, Carlsbad, CA, USA) and consisted of 40 cycles with denaturation at 95 °C for 5 s, annealing at 60 °C for 10 s, and elongation at 60 °C for 60 s. All experimental conditions were assessed in triplicates. The primers were used as follows: *TACR1*-*tr* forward, 5′-CAGGGGCCACAAGACCATCTA-3′; *TACR1-tr* reverse, 5′-ATAAGTTAGCTGCAGTCCCCAC-3′; *TACR1-fl* forward, 5′-AACCCCATCATCTACTGCTGC-3′; *TACR1-fl* reverse, 5′-ATTTCCAGCCCCTCATAGTCG-3′; TAC1 forward, 5′-TCGTGGCCTTGGCAGTCTTT-3′; *TAC1* reverse, 5′-CTGGTCGCTGTCGTACCAGT-3, *GAPDH* forward, 5′-GTCTCCTCTGACTTCAACAGC-3′; *GAPDH* reverse, 5’-ACCACCCTGTTGCTGTAGCCAA-3’. *ZEB1* forward, 5′-TTCACAGTGGAGAGAAGCCA-3′; *ZEB1* reverse, 5′-GCCTGGTGATGCTGAAAGAG-3′; *CDH1* forward, 5′-GAACGCATTGCCACATACAC-3′; *CDH1* reverse, 5′-ATTCGGGCTTGTTGTCATTC-3′. All kits were used according to the manufacturer’s instructions. Primer functionality was confirmed in the hepatoblastoma cell line Hep G2 for *TACR1-tr*, *TACR1-fl*, and *TAC1*. Hep G2 cDNA was obtained from Kolorz et al., 2021 [24]. 

### 2.7. Apoptosis Detection Assay

Cells were seeded at 500,000 cells per well in a 6-well plate and incubated for 24 h at 37 °C, 5% CO_2_. The treatments, aprepitant (Tocris Bioscience, Bristol, UK) and substance P (Tocris Bioscience, Bristol, UK), were applied to the wells at their respective concentrations and incubated for 24 h at 37 °C, 5% CO_2_. Post treatment, the cells were observed for morphological changes under the microscope. The supernatants from each well were individually collected into 5 mL round bottom polystyrene test tubes (Falcon, Corning, Wiesbaden, Germany). The cells were carefully washed with 1 mL of DPBS (PAN-Biotech, Aidenbach, Germany) per well. The cells were then detached using 300 μL of accutase (Sigma-Aldrich, Taufkirchen, Germany) and incubated for 3 min at 37 °C, 5% CO_2_. Accutase was inactivated by adding 1.5 mL of medium with 10% FBS (Falcon, Corning, Wiesbaden, Germany). The cells were carefully collected into their respective FACS tubes and centrifuged (Hettich, Rotina 380R) at 500 rpm for 5 min. The pellet was resuspended in 1 mL of DPBS and the centrifugation step was repeated. The cell pellet was stained for 15 min at 37 °C, 5% CO_2_ with 100 μL of working solution (96 μL DPBS + 3 µL Annexin V + 1 μL propidium iodide (FITC Annexin V Apoptosis Detection Kit I, BD Biosciences, Heidelberg, Germany)). The staining was stopped by washing cells with 900 µL 1× ABB. The cells were measured using flow cytometry (LSRFortessa, BD Biosciences, Heidelberg, Germany). Further analysis was performed using FlowJo software (BD Biosciences, version 10).

### 2.8. Caspase Detection Assay

Cells were seeded at 500,000 cells per well in a 6-well plate and incubated for 24 h at 37 °C, 5% CO_2_. The treatments, aprepitant (Tocris Bioscience, Bristol, UK) and substance P (Tocris Bioscience, Bristol, UK), were applied as indicated. Post treatment, the cells were observed for morphological changes under the microscope. The supernatants from each well were individually collected into 5 mL round-bottom polystyrene test tubes (Falcon, Corning, Wiesbaden, Germany). The cells were carefully washed with 1 mL of DPBS (PAN-Biotech, Aidenbach, Germany) per well. The cells were then detached using 500 µL of trypsin (PAN-Biotech, Aidenbach, Germany) and incubated for 3 min at 37 °C, 5% CO_2_. The trypsinization was stopped by adding 1.5 mL of medium with 10% FBS. The cells were carefully collected into 5 mL round-bottom polystyrene test tubes (Falcon, Corning, Wiesbaden, Germany) and centrifuged at 500 rpm for 5 min. The cell pellet was re-suspended in 1 mL of DPBS and the centrifugation step was repeated. The cell pellet was stained for 30 min at 37 °C, 5% CO_2_ with 100 µL of working solution (99 µL DPBS + 5% FBS + 1 µL FITC–VAD-FMK (Apostat intracellular caspase detection, R&D Systems, Minneapolis, MN, USA). The staining was stopped by washing cells with 1 mL DPBS and centrifugation. The cell pellet was re-suspended in 1 mL DPBS and analyzed using flow cytometry. Further analysis was performed using Flowjo software (version 10) by BD.

### 2.9. Cell Cycle Detection

Cells were seeded at 500,000 cells per well in a 6-well plate and incubated for 24 h at 37 °C, 5% CO_2_. The treatments, aprepitant and substance P, were applied to the wells at their respective concentrations and incubated for 24 h. The supernatants from each well were individually collected into 5 mL round-bottom polystyrene test tubes. The cells were carefully washed with 1 mL of DPBS per well. The cells were then detached using 500 µL of trypsin and incubated for 3 min at 37 °C, 5% CO_2_. The trypsin was inactivated by adding 1.5 mL of medium with 10% FBS. The cells were carefully collected into their respective FACS tubes and centrifuged at 500 rpm for 5 min. The pellet was resuspended in 1 mL of DPBS and the centrifugation step was repeated. The cells were fixed by dropwise addition of 70% ice cold ethanol. The cells were incubated overnight at 4 °C to aid fixation. The cells were then centrifuged to remove the traces of ethanol and washed repeatedly with 1 ml DPBS. The cell pellets were stained with 1 mL 4′,6-Diamidin-2-phenylindol, Dihydrochlorid (DAPI) working solution (1 µg/mL DAPI (Invitrogen, Carlsbad, CA, USA) + 1% Triton X (Merck, Darmstadt, Germany) in DPBS) for 15 min in the dark. The cells were analyzed using flow cytometry. Further analysis was performed using FlowJo software (version 10) by BD Biosciences (San Jose, CA, USA).

### 2.10. ELISA

Conditioned media from the cell lines, MIA Paca-2, Panc-1, DanG, and HuP-T3, were collected after 24 h of culture to evaluate them for residual substance P concentration. The study was also extended to include sera from cancer patients and from control groups. This was approved by the ethics committee of the Ludwig-Maximilian-University (LMU) Munich, Germany (approval number 19–233). The supernatants were centrifuged at 4 °C, 16,000× *g* for 10 min. An amount of 100 µL of the test samples were applied to a pre-coated plate and the further steps were performed as recommended by the manufacturer (Substance P ELISA Kit, My BioSource, San Diego, CA, USA). The absorbance was detected at 450 nm using a spectrophotometer (VersaMax, Molecular Diagnostics, CA, USA). The minimum detectable concentration of SP was 0.175 ng/mL. 

### 2.11. Statistical Analysis

Results are expressed as the mean ± standard error of the mean (SEM). All statistical comparisons were performed via one way ANOVA or unpaired parametric *t*-test comparing two groups using the biostatistics software GraphPad Prism (version 9.0.0, 86, San Diego, CA, USA). *p*-values shown as *, *p* ≤ 0.05; **, *p* ≤ 0.01; *** *p* ≤ 0.001; ****, *p* ≤ 0.0001. 

## 3. Results

### 3.1. Transcriptome Data Analysis Reveals Diverging Expression of TACR1 and TAC1 in PDAC and Normal Pancreatic Tissue

To obtain an overview regarding the importance of the SP/NK1R-complex in PDAC, we first conducted expression analysis of complex-related genes via RT-qPCR. The assay was performed on eight PDAC cell lines as indicated in Figure 1a, targeting both NK1R isoforms (*TACR1-tr* and *-fl*) separately, as well as *TAC1*. Additionally, we used the same for expression analysis on primary stellate cells. While no expression of *TACR1-fl* was detected consistently throughout all RT-qPCR experiments, we observed fluctuations in *TACR1-tr* expression between the pancreatic cancer cells. No expression could be detected in Capan-1 and HuP-T3, however, we observed *TACR1-tr* in the remaining six PDAC cell lines and, with low levels, in PSCs. In comparison to the positive control (hepatoblastoma cell line Hep G2) [20], we found significantly lower *TACR1-tr* levels in all tested PDAC cells (*p*-value < 0.0001). Although differences in *TAC1* expression were not within a significant range, it is noteworthy that Panc-1, MIA PaCa-2, Hep G2, and PSCs were positive for the presence of the SP encoding gene, whereas all other PDAC cell lines were categorized as non-detects (Figure 1a). 

In contrast to the RT-qPCR results, quantification of SP employing Enzyme-Linked Immunosorbent Assay (ELISA) in cell lines did show SP presence in all tested cell lines, with highest levels also in MIA PaCa-2 (Figure 1b). We performed the same assay on human serum for comparison of SP levels between control and PDAC patients. Here, we found a lower SP blood serum level in PDAC patients in comparison to a control group, although not with a significant level (Figure 1c). 

Next, we collected publicly available bioinformatical data sets for further validation of SP/NK1R complex related gene expressions. The comparison of the PDAC data sets obtained from Gene Expression Omnibus (GEO), the Cancer Genome Atlas (TCGA), and Cancer Cell Line Encyclopedia (CCLE) allowed identification of a significant downregulation of *TACR1* in tumor vs. normal cells (Figure 2a). However, these public records did not allow for the differentiation between the two receptor splice variants. 

Interestingly, GEO data indicates differential gene expression with tumor progression. Precisely, GSE micro array data shows tendencies towards a decreasing expression of total *TACR1* with tumor stage progression, while *TAC1* tends to increase with higher stages (Figure 2b). In addition, in survival analysis, higher expressions of *TACR1* correlated significantly with higher overall survival for PDAC patients (Figure 2c, via OncoLnc.org).

### 3.2. Epithelial to Mesenchymal Transition State Correlates with Expression of TACR1 and TAC1

Epithelial-mesenchymal plasticity (EMP) describes the disruption of tissue homeostasis, which further contributes to cellular transformation and heterogeneity, particularly in PDAC [25]. Recent biological experimental data, as well as transcriptomic bioinformatical analysis, support the strong association between high expression of mesenchymal markers, such as the zinc finger E-box binding homeobox 1 (*ZEB1*), and worse prognosis in patients [26]. Interestingly, inverse correlation of gene expression of *TACR1-total* and *ZEB1* demonstrates a significant relationship between high expression of *ZEB1* and low expression of *TACR1-total* (Figure 2d). However, we found a positive correlation between *ZEB1* and *TAC1* expression (Figure 2e). Gene clustering revealed *TACR1* and *TAC1* to form two distinct clusters, each with a group of different EMT markers. Hereby, *TACR1* clustered with genes associated with epithelial characteristics (e.g., *S100A7A*, *SNAI2*, *KRT19*, *CDH1*), while *TAC1* clustered closely to mesenchymal markers (e.g., *ZEB1*, *CDH2*, *VIM*, *ZEB2*) corroborating before mentioned data. Representing the heterogenic nature of PDAC for our further investigations, we used PDAC cell lines (Capan-1, DanG, HuP-T3, Panc-1, and MIA PaCa-2) that differ in their classification regarding epithelial-to-mesenchymal-transition (EMT) state as well as based on their *TACR1* and *TAC1* gene expression according to the CCLE data base (Figure 1a and Figure 2f, and Table 1). To provide an overview on those cell line characteristics relevant to this study, Table 1 shows the EMT state of eight cell lines as well as a simplified classification model of gene expression based on CCLE data (see also Figure 2f).

To validate EMT state and expression levels of EMT markers, we employed primers for *ZEB1* and epithelial cadherin-1 *(CDH1*), additionally to the aforementioned primer sets in all RT-qPCR runs. We found the expression of EMT markers in our data to match the indicated classifications in previous literature [27]. In more detail, *CDH1* expression in DanG was significantly higher in comparison to the other PDAC cell lines with no expression in MIA PaCa-2 and PSN-1. *ZEB1* was detected in BxPC-3, AsPC-1, PSN-1, DanG, HuP-T3, Panc-1, Hep G2 (sorted in ascending order), and, with the significantly highest expression, in MIA PaCa-2 (Figure 1a). 

### 3.3. Aprepitant Significantly Reduces Growth in PDAC Cell Lines and Cancer Stem Cell-Like Cells

To investigate the effects of NK1R-targeted therapy on the growth of PDAC cells, we tested treatment with the NK1R antagonist AP followed by the determination of 50% inhibitory concentration by MTT cell viability assay. The IC50 value for Hep G2, obtained by Kolorz et al., 2021 [24], functioned as positive control [20]. Exposure to AP resulted in dose-dependent growth inhibition of PDAC cells over a time period of 24 h. The lowest sensitivity to AP was detected in PSCs (32 µM), followed by Panc-1 (30 µM), Capan-1 (30 µM), HuP-T3 (29 µM), DanG (26 µM), and MIA PaCa-2 (19 µM) (Figure 3a). Similar IC50 values have been previously detected [28].

PSC-conditioned media is known to fuel pancreatic cancer cell metabolism and stimulate tumor cell proliferation and colony formation [29]. To exclude potential discrepancies through potential SP secretions of the immediate PDAC microenvironment represented by PSCs, we determined IC50 values (MTT) for both normal and PSC conditioned media for all cell lines. Interestingly, we observed higher susceptibility to AP treatment in cells cultured in PSC-conditioned media (Figure 3b). A significant difference between the two growth conditions was not ascertainable (unpaired *t*-test). 

An in vitro surrogate functional marker and technique to identify cell lines enriched of cells with stem-like characteristics is the establishment of colonies and spheroids [23,30,31]. Treatment of colonies and spheres resulted in exceptional dose-dependent treatment response to AP after 14 days of culture. In the presence of SP, effects were partially smoothened, especially in co-treatments with higher concentrations of AP, particularly in MIA PaCa-2. In addition, MIA PaCa-2 robustly showed the highest sensitivity to AP treatment. Remarkably, we found a profound and significant inability of colony and spheroid formation at a concentration of 40 µM consistently in all cell lines (*p* < 0.0001). Addition of SP seemed to profit Panc-1 as the only cell line under CFA growth conditions (*p* = 0.0005), whereas effects of AP seemed to be alleviated by additions of SP in SFAs with MIA PaCa-2. There was no significant difference between control and SP-only treatment in SFA (Figure 3c).

In addition to changes in the number of colonies and spheroids, we also observed morphological differences after the treatments. Figure 3d illustrates spheroids obtained from DanG, MIA PaCa-2, and Panc-1. Distinct phenotypic features, including size, shape, and texture were monitored. Specifically, manipulation of the SP/NK1R system led to a loss of tightly packed shape in all cases, with blebbing at the surface of the spheres indicative of apoptosis initiation after antagonistic treatment (Figure 3d, right panels). 

In summary, we observed a dose-dependent decrease in PDAC cell viability reflected by reductions in the size of spheres and numbers of cells as well as changes in cell morphology after exposure to the NK1R antagonist AP. 

### 3.4. Aprepitant Affects Cell Cycle Progression in PDAC Cells

We performed FITC Annexin V/PI staining to measure treatment-induced apoptosis rates (Figure 4a) as well as pan caspase labeling for determination of caspase activity via flow cytometry (Figure 4b). For apoptosis detection, our experiments revealed no differences between the treatment regimens in all cell lines. Moreover, caspase detection solely indicated a slight left shift in MIA PaCa-2, suggesting low pan caspase contribution with respect to apoptosis (Figure 4b). On the basis of these results, we performed DAPI staining for flow cytometry analysis to explore cell cycle progression in treated vs. untreated PDAC cell lines. The collected data showed drastic AP-induced changes of events in the phases G1 and S in the three cell lines DanG, Panc-1, and MIA PaCa-2 comparing AP-treated cells to controls. However, Capan-1 and HuP-T3, which had no expression of *TACR1-tr* and *-fl*, showed no reaction regarding AP-induced cell cycle arrest. We observed no effect of SP in terms of cell cycle modulation in all cells. For more detailed information, we added the histograms illustrating changes in cell cycle progression after AP treatment into the appendix. Taken together, blockage of NK1R in *TACR1-tr* expressing PDAC cells lead to prominent cell cycle arrest with an emphasis on G1 and S phase (Figure 5).

## 4. Discussion

The SP/NK1R complex, especially the involvement of NK1R-tr, is known to play a pivotal role in different solid cancer entities, such as the childhood cancers hepatoblastoma (HB) and neuroblastoma as well as colon and breast cancer. Our latest publications firstly demonstrated the robust efficacy of the NK1R antagonist AP concerning the inhibition of pediatric liver cancer in vitro and in vivo [20]. Secondly, we demonstrated AP-induced modulation of AKT and Wnt pathways for the same cancer type [22]. Similarly, we reported the Wnt/β-catenin signaling activity to be significantly inhibited in colorectal cancer cells, thus decreasing cancer stemness [21]. To expand on these findings, in this study, we investigated the role of the SP/NK1R complex in the tumorigenesis of PDAC with a focus on the potential anti-tumoral effects of the NK1R antagonist AP. 

A number of studies has already investigated splice variant subordinate differences between normal and malignant tissues, proving the homogenic and tissue specific presence of either one of the two isoforms [14,20,32]. Whereas the full transcript was commonly identified in areas of the central and peripheral nervous system, the truncated form is expressed in several tissues and cells [33]. We hypothesized PDAC cells to predominantly express NK1R-tr over the full-length version, which was confirmed through RT-qPCR analysis, while also providing proof for the absence of NK1R-fl in all tested PDAC cells. 

In contrast to our results suggesting no expression of *TACR1* in some cell lines, such as Capan-1, other publications have reported contradictory results in Capan-1 via RT-qPCR and Western blot [34,35]. Depending on the method employed, RT-qPCR performance can yield different results and thus show discrepancies between the measures of standard error [36]. Additionally, Friess et al. (2003) [35] employed a primer set binding to the first exon of the gene, thus detecting both variants at the same time. In contrast, our primers were designed to specifically distinguish between the NK1R isoforms [TACR1-fl,20], thus, a decrease in expression might be expected compared to mRNA of total NK1R (full and truncated combined). However, the absence of the gene on a transcriptional level might also be explained through features associated with the truncated isoform of NK1R. In contrast to NK1R-fl, following endocytosis, the carboxy-terminally truncated, and thus internalization-defective, NK1R-tr fails β-arrestin-mediated dephosphorylation. This can cause unsuccessful desensitization of NK1R-tr to SP [11,16]. The prolonged exposure of NK1R to substance P has been shown to lead to the decrease of NK1R on a transcriptional level with concurrent increase of regulatory microRNAs (miRNA) [37]. As such, the presence of the protein might also explain the strong AP-induced effects we observed in our experiments despite the here-observed low or absent transcriptional expression. Furthermore, variations between transcriptional and protein levels are a phenomenon frequently observed [38,39]. For further investigation and determination of gene expression, we suggest Western blot analysis for the specific protein of interest. However, with respect to this method, it seems to be difficult to distinguish between the two isoforms so far.

Varying expression of *TACR1-tr* and *TAC1* release is in line with the TCGA data, where we found trends related to tumor stages. Such intermittent transcriptomic expression could be required for tumor stage development; however, definite assertions require further investigation. Another explanation that needs to be considered, is the downregulation of *TACR1* due to genomic alterations with progressing EMT [27]. As such, cells might try to counterbalance the loss of function of NK1R by upregulating its activator SP on a transcriptomic level. In a first step to understand circulating SP serum levels in patients, we screened SP levels in PDAC patients in comparison to patients without an underlying malignancy and found a slight tendency towards a decrease of SP serum levels in the former. However, to increase the predictive significance of this data, a higher number of samples would be required to determine differences in SP serum levels. Our data let us assume that local transcriptomic data of *TAC1* and circulating protein release measurements in serum do not necessarily correlate and might therefore not be helpful for monitoring of PDAC. Therapeutic stratification or monitoring of therapeutic success with NK1R inhibition will have to be investigated in future trials.

Data mining in publicly available datasets, such as TCGA, GEO, and CCLE, with additional consideration of OncoLnc survival data, allowed for linkage of differential expression of *TACR1* and *TAC1* to tumor progression and EMT state of cancer cells. EMT is an act of tumor progression, in which epithelial cells lose their cell polarity and cell-cell adhesion in order to gain migratory and invasive properties to become more mesenchymal [25,40,41]. Significant decline of *TACR1* in PDAC tumor tissues in the course of tumor progression is in line with survival curves demonstrating lower expression of *TACR1* to correlate with poorer survival. On top of that, low expression of *TACR1* in PDAC (expression compared to other tissues) significantly relates to high expression of *ZEB1* (Figure 2d). *ZEB1* overexpression is known to facilitate tumor progression, invasion, and metastasis [40]. In summary, exponential decline of *TACR1* seems to correlate with exponential increase of *ZEB1* expression. Clustering of gene expression revealed grouping of *TACR1* and *TAC1* into two distinct sets. More precisely, *TACR1* showed closer relation to epithelial markers, whereas *TAC1* was clustered into a group of genes associated with mesenchymal properties. Positive correlation of *ZEB1* and *TAC1* also showed high levels of these genes to correlate with a close to significant *p*-value. It remains particularly intriguing, whether the transition of high *TACR1* and low *TAC1* might act as an indicator of tumor progression, especially as links between substance P and activation of mesenchymal stem cells were previously described [42]. However, the presented results require biological conformation as being based solely on bioinformatical analysis. At this point, we do not consider *TAC1* as a molecular marker for PDAC diagnosis, but we hypothesize that the decrease of SP in patient sera might be caused by increased SP consumption by malignant cells and might be an object for further exploration.

In a previous study [35], mRNA expression of NK1R in pancreatic tissue was analyzed via RT-qPCR. According to these results, gene expression indicates higher expression of NK1R in tumor versus normal tissue. Additionally, they found increasing NK1R levels with tumor progression, as well as higher expression of NK1R to correlate with lower patient survival. Surprisingly, the data sets analyzed in this study (Figure 2; TCGA, GSE62165, GSE15471, and GSE16515) clearly show contradictory results. Although both results are based on transcriptomic analysis, contradictory results might still occur through differences in the methodology, as GEO data sets for instance are microarray-based. Furthermore, bioinformatical analysis conducted in our study employed four large independent data sets with a total of 193 tumor patients and 68 normal controls in the GEO data set, and another 86 patients through OncoLnc. Thus, discrepancies might occur through a much smaller population size in the aforementioned publication. 

High expression of truncated *TACR1* and high sensitivity to AP seem to correlate. Interestingly, we found strikingly higher expressions of truncated *TACR1* in MIA PaCa-2, which also exhibited the highest values regarding the presence of *TAC1*. Furthermore, we found the mesenchymal cell line to be the most sensitive for AP treatment in terms of growth interference in all experiments, including CFA and SFA. Additionally, we observed a small increase in pan caspase activity solely in this cell line, suggesting AP to have a particular effect on this strongly undifferentiated cancer cell. Similar trends observed in MIA PaCa-2 were found in DanG, which showed the second highest level of *TACR1-tr*. Taken together, our investigation suggests cancer subgroups with higher expression of *TACR1-tr* to be more susceptible for AP. For future studies, we therefore believe that PDAC patients with higher *TACR1-tr* expression might potentially benefit from treatment with AP as an anti-cancer treatment. Hence, we propose a prospective trial to investigate the implementation of *TACR1-tr* measurement for therapeutic stratification of PDAC.

The establishment of tumor-derived spheroids from anoikis-resistant cells is commonly used as an in vitro surrogate functional marker for cancer stemness [23]. It enables for sensitive identification of cells possessing CSC-like properties while at the same time allowing drug testing in a 3D in vitro culturing model [23,30,31]. CSCs are rare tumor initiators with strong chemoresistance [30,31]. The potential to inhibit CSC growth is therefore considered a very attractive method to improve therapy effects in cancer patients. Not only did most of the here-tested cell lines show the ability for CFA and SFA, but also, we observed exceptional dose-dependent treatment responses to AP in colonies and spheroids after 14 days of culture. In addition, cells in colony and spheroid formation exhibited even higher sensitivity to AP than parental cells, indicating lower concentrations to be required for successful growth inhibition. This suggests NK1R blockage to show high efficacy in targeting CSC-like cells, thus owning high potential for tumor initiator inhibition. In order to gain more knowledge regarding this mechanism, further analysis is required at this point to determine potential changes in the expression of SP/NK1R-related genes. 

Another crucial discovery is the effect of AP on PSCs, where we found a lower sensitivity. PSCs are the major contributor to aggressive stromal fibrosis and closely interact with cancer cells, which in turn stimulates pancreatic tumor growth [43,44]. Activation of quiescent PSCs during PDAC development promotes several factors associated with proliferation and tumor progression. Additionally, linkage to genomic instability and capability of induction of EMT has been reported [44]. The ability to silence cell signaling of such cells might drastically improve a patient’s outcome through inhibiting reoccurring tumor growth after surgery. 

Due to the markedly decreased viability in all assays after AP exposure, we examined the cells regarding AP-induced mechanisms. Intriguingly, we found very low indication for activated apoptotic processes. With both apoptosis assays uniformly suggesting the exclusion of apoptotic mechanisms in AP-induced growth inhibition, we refocused on a different mechanism to determine the effect of NK1R antagonistic growth inhibition. One way to eliminate cancer cells is interference with cell cycle progression, the key process for cell replication. Quantification of cell cycle progression allowed the identification of AP-induced cell cycle arrest with the most striking effects in G1 and S phase. To further uncover the mechanisms behind AP-induced cell cycle arrest, deeper investigations are of priority including, among others, cell-cycle-related proteins.

Relative overall mRNA expression of *TACR1* splice variants led us to question the mechanism through which AP exhibits its significant anti-proliferative effect on pancreatic cancer cells, which we expected to be mediated by the SP/NK1R complex. Next to binding to NK1R with high affinity, AP also shows little to no affinity to corticosteroid receptors, serotonin, or dopamine [45]. For further investigation, it might be of interest whether aprepitant could exhibit anti-tumoral effects through binding to the mentioned receptors in case of NK1R absence. 

## 5. Conclusions

Current research strongly supports the idea of the SP/NK1R complex being involved in cancer progression. Additionally, it has been shown in multiple ways that blockage of the NK1R receptor results in the inhibition of cancer cell growth. In this study, we identified the SP/NK1R complex as a potent target in PDAC and aprepitant to inhibit cell cycle progression. NK1R blockage resulted in dose-dependent growth reduction in CSC-like cells, parental PDAC cells, and to a lesser degree in primary PSCs, whereby the highest sensitivity was observed in aggressive cancer cell types and subgroups expressing higher levels of the truncated *TACR1* variant. *TACR1-tr* was also the predominantly expressed isoform in PDAC cells. In conclusion, we suggest antagonistic NK1R-blockage as a potential therapeutic option for PDAC subgroups with high *TACR1-tr* expressions.

However, with respect to the heterogenic nature of PDAC, deeper investigation of splice variants appearing in PDAC patients is necessary and might help to distinguish between PDAC subgroups. Especially with the currently increasing identification of isoform-guided differential mechanisms being prominently involved in tumor progression, increased understanding might be gained through public availability of transcriptomic differentiation between splice variants. So far, only a very small number of studies has investigated the effects of AP in cancer patients on a molecular level regarding its role as a potential cancer drug. Emphasizing on PDAC being a very heterogenic tumor with no dominant druggable mutation [46,47], investigating the mechanisms on a transcriptional level has high potential in highlighting crucial differences between tumor subtypes and their underlying mechanisms in tumor progression. With no substantial improvements in PDAC treatments over the past 30 years [48,49], such information has the power to lead to new therapeutic developments. 

## Figures and Tables

**Figure 1 cancers-13-02703-f001:**
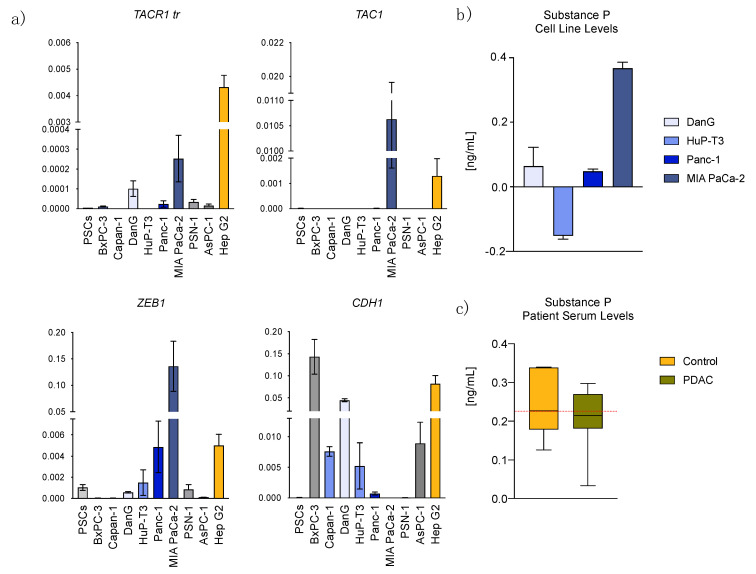
Expression of the SP/NK1R complex varies in established cell lines and PDAC patient serum. (**a**) Transcript abundance analyzed through RT-qPCR illustrated in ∆Ct with standard deviation (SD) in the indicated cell lines. (**b**) ELISA for quantitative analysis of Substance P levels in PDAC cell lines and (**c**) serum of control (*n* = 7) vs. PDAC patients (*n* = 7).

**Figure 2 cancers-13-02703-f002:**
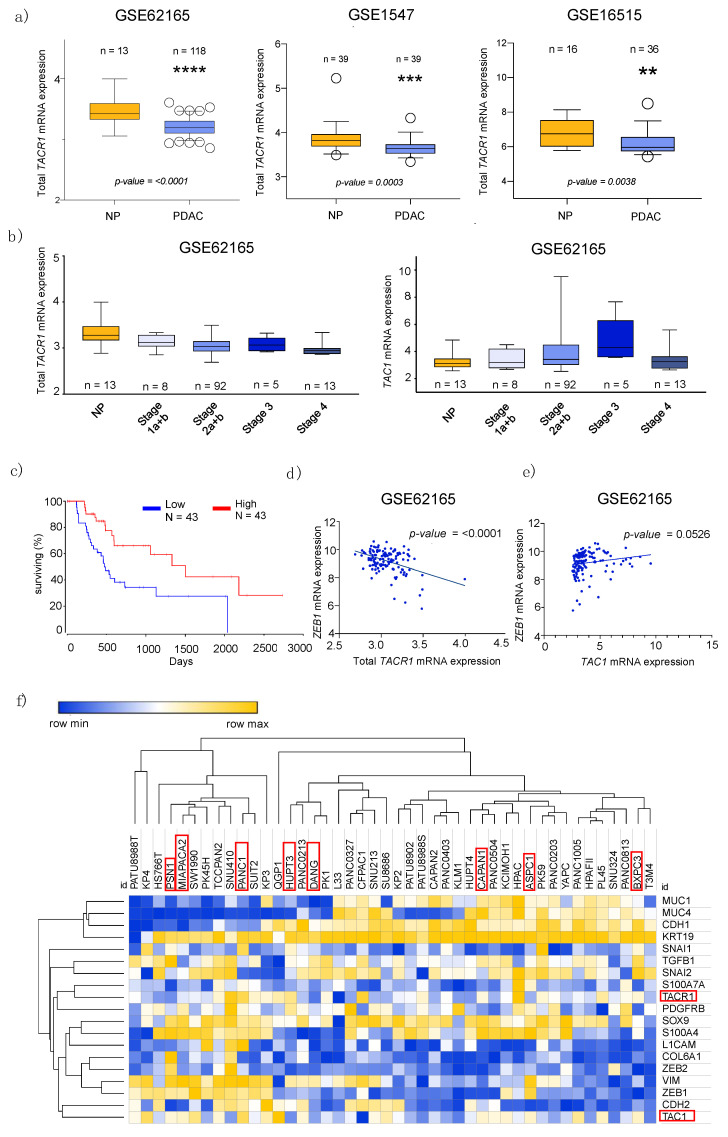
Transcriptome-based data analysis. (**a**) Boxplots demonstrating significantly lower mRNA expression levels of *TACR1* in PDAC compared to normal tissue (NP) in the indicated transcriptomic data series. *p*-values were calculated using unpaired *t*-test. (**b**) *TACR1-total* and *TAC1* gene expression in different pancreatic cancer stages obtained from the indicated GEO data set. (**c**) Prognostic significance of *TACR1-tr* (top and bottom 25% gene expression) in PDAC, assessed by OncoLnc. (**d**) Inverse correlation between the EMT marker *ZEB1* and *TACR1* in the indicated transcriptomic data set. The *p*-value is based on Pearson’s Correlation. (**e**) Correlation between *ZEB1* and *TAC1* in the indicated transcriptomic data set. (**f**) Visualization of CCLE gene expression across PDAC cell lines. The SP/NK1R coding genes are highlighted in red, as well as the cell lines used for further analysis. Hierarchical clustering based on one minus Pearson’s Correlation.

**Figure 3 cancers-13-02703-f003:**
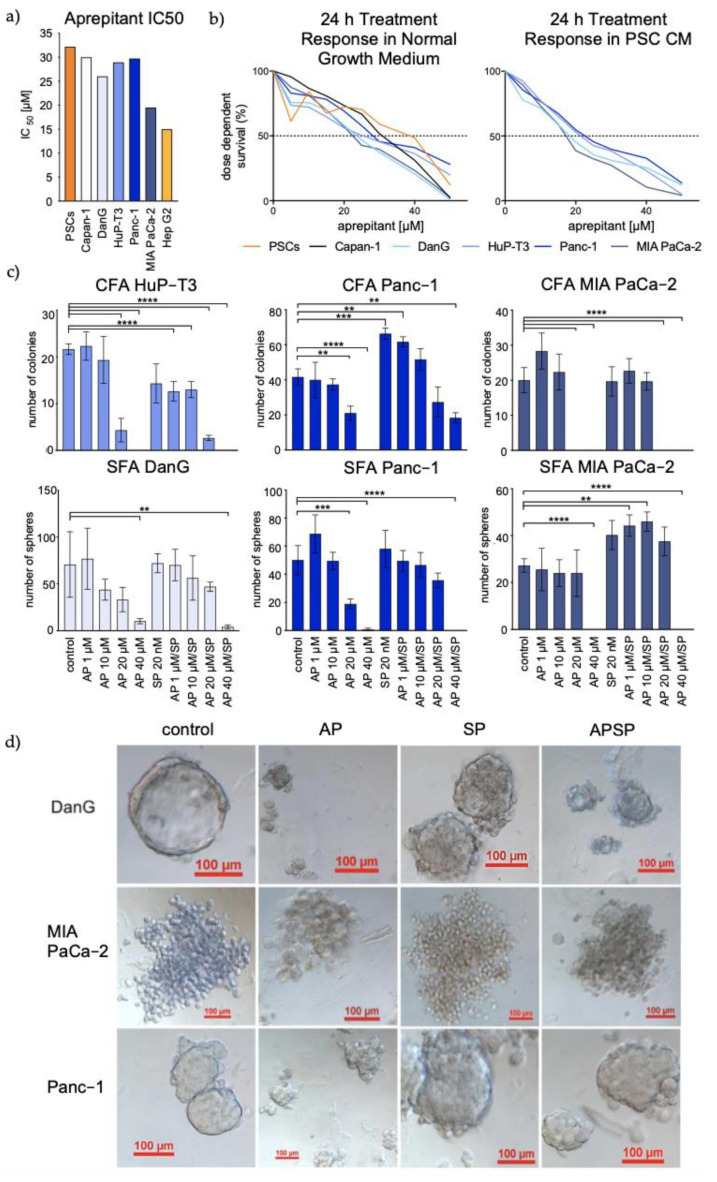
Aprepitant treatment response in PDAC cells, spheres, and colonies. (**a**) IC50 for all tested PDAC cell types as indicated. (**b**) AP dose-dependent survival curves for the indicated PDAC cells in normal (left) or PSC-conditioned media (right). (**c**) Bar charts illustrating number of colonies (upper panel) or spheroids (lower panel) under different treatment conditions as specified on the x-axis. (**d**) Images of SFA under treatment conditions showing morphological changes in the cell lines DanG (upper panel), MIA PaCa-2 (middle panel), and Panc-1 (lower panel).

**Figure 4 cancers-13-02703-f004:**
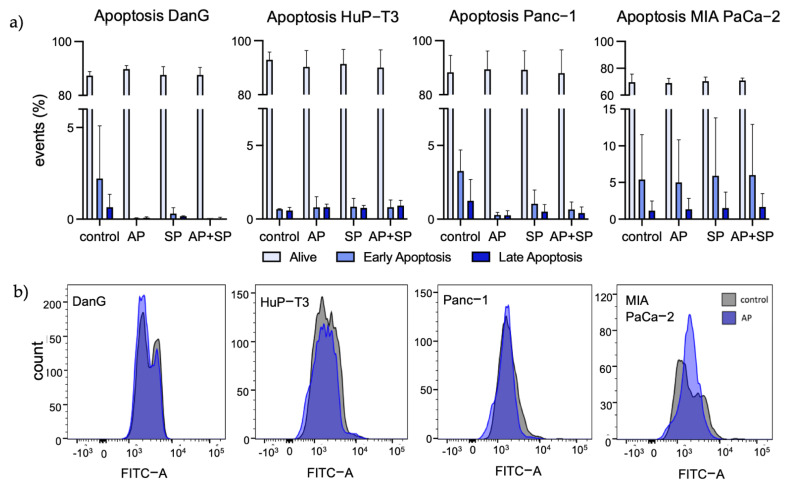
Apoptosis detection and caspase detection analysis. (**a**) FITC Annexin V/PI flow cytometry for detection of AP-mediated apoptosis in untreated and SP-treated cells. Drugs were applied in the following concentrations: AP 25 µM and SP 20 nM. Statistical analysis revealed no significant differences between treatments regarding Annexin V-positive cells. (**b**) FITC-VAD-FMK accumulation in the indicated cell lines measured via flow cytometry. Slight left shift after AP-treatment indicates small, but not significant changes in intracellular caspase activation. Bar charts and univariate histogram sorted left to right from epithelial to mesenchymal transition state.

**Figure 5 cancers-13-02703-f005:**
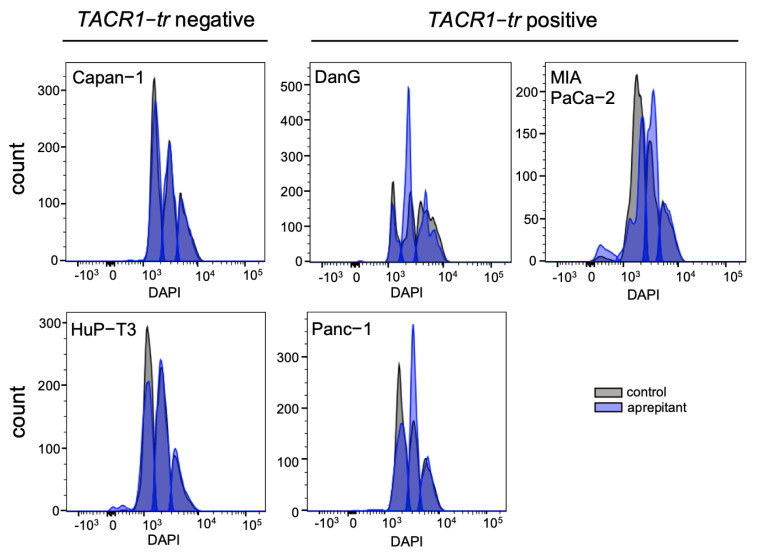
Evaluation of cell cycle progression in PDAC cells. Flow cytometry univariate histogram of DAPI staining in control vs. AP-treated cells showing number of events for cell cycle phases. Left panels: *NK1R-tr* negative cell lines demonstrate no difference in treatment. Right panels: a clear shift in cell cycle progression could be observed in *NK1R-tr* positive cell lines. The histogram of cell cycle distribution was generated from 10,000 events per sample. Appendix A provide more detailed information regarding the AP-induced shifts in cell cycle progression. Drugs were applied in the concentrations 25 µM of AP and 20 nM of SP.

**Table 1 cancers-13-02703-t001:** PDAC cell line characteristics and CCLE gene expressions.

PDAC Cell Line	EMT State	*TAC1*	*TACR1*
BxPC-3	epithelial	+	+
Capan-1	epithelial	+	+
DanG	epithelial	− −	− −
HuP-T3	epithelial/mesenchymal	− −	+ +
Panc-1	epithelial/mesenchymal	+ +	+ +
MIA PaCa-2	mesenchymal	+	−
PSN-1	mesenchymal	+	−
AsPC-1	mesenchymal	+	+

− − = very low expression; − = low expression; + = high expression; + + = very high expression.

## Data Availability

TCGA at https://portal.gdc.cancer.gov, last accessed on 6 March 2021; GEO at https://www.ncbi.nlm.nih.gov/geo/, last accessed on 6 March 2021; CCLE at https://portals.broadinstitute.org/ccle, last accessed on the 14 April 2021; OncoLnc at http://www.oncolnc.org, last accessed on the 14 April 2021.

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
