# Peer review of "Identification of the Neurokinin-1 Receptor as Targetable Stratification Factor for Drug Repurposing in Pancreatic Cancer"

_cancers, 2021, doi:10.3390/cancers13112703_

Round 1

Reviewer 1 Report

This study investigated the correlation of NK-1R/SP gene with cancer cell growth, using diver cancer cell lines in vitro. In order to explore the finding, author tried to analyze TARC1/TAC1 level and SP concentration in cell line and serum of patients. The blockage of NK-1R affects cell cycle without apoptosis, leading to low growth rate. This finding is very important and provides the critical information to the clinical application.

  • Author provided the sufficient data to support the hypothesis but some data is lack of statistical analysis and error bar. Please represent the data with the statistical meaning.
  • AP treatment affected shape of spheroid/colony as well as its number in Figure 3. How about AP+SP-treated condition?
  • How can author determine SP concentration used in this study?
  • Author identified expression of TARC1/TAC1 in cell line and referred to other reports. However, considering the data from this study, the correlation between TARC1/TAC1 and clinical outcome is difficult to be confident. Explanation had better be revised.
  • Author revealed that TAC1/TACR1 expression is different from cell type; mesenchymal and epithelial cell type as shown in table 1. However, table 1 may not show the trend of TAC1/TACR1 expression depending on cell characteristics. This requires experiment with more cell lines.

Author Response

We thank you for the careful review and valuable comments and suggestions. For a detailed point-by-point response, please see the attached PDF with our answers held in blue.

Reviewer 2 Report

It is an interesting original research about identification of the Neurokinin-1 Receptor as targetable stratification factor for drug repurposing in Pancreatic Cancer. Targeting of the NK-1R with its antagonist aprepitant resulted in anti-tumor effects in pancreatic cancer. They treated human pancreatic ductal adenocarcinoma (PDAC) cell lines (Capan-1, DanG, HuP-T3, Panc-1, and MIA PaCa-2) and their cancer stem cell-like cells (CSCs) with aprepitant. The treatment with NK-1R antagonist aprepitant resulted in dose-dependent growth reduction in both CSCs and non-CSCs without induction of apoptosis in most PDAC cell lines.

However, there are some important issues

Major points

- It is known that tumor tissues and cells overexpress more NK-1R than normal tissues and cells. In addition, SP induces tumor cell proliferation. The SP mitogenic action is carried out via the NK-1R, since the growth inhibition of many human tumor cells after the administration of NK-1R antagonists is partially reversed by the administration of SP (Peptides 2013, 48, 1-9). Moreover, it has been reported by real-time quantitative RT-PCR, NK-1R mRNA was increased 36.7-fold (p < 0.001) in human pancreatic cancer samples compared with normal controls. By Western blot analysis, NK-1R was increased 26-fold (p < 0.01) in pancreatic cancer samples in comparison to normal controls.  NK-1R mRNA was detected in five pancreatic cancer cell lines (Panc-1, MIA-PaCa-2, ASPC-1, Capan-1, and T3M4) by real-time quantitative RT-PCR, with the highest levels in CAPAN-1 cells and the lowest in ASPC-1 cells (Lab Investigation 2003, 83, 731-742). However, the authors no detected NK-1R mRNA in five pancreatic cancer cell lines (Capan-1, DanG, HuP-T3, Panc-1, and MIA PaCa-2). As can be seen, there are three same pancreatic cancer cell lines (Panc-1, MIA-PaCa-2 and Capan-1) studied by the two groups. However, NK-1R mRNA was detected in (Panc-1, MIA-PaCa-2 and Capan-1) with the highest levels in Capan-1 (Lab Investigation 2003, 83, 731-742) and you no detected NK-1R mRNA in these cells. This should be discussed. In addition, the papers above mentioned should be included and discussed.

- Enhanced NK-1R expression levels were associated with advanced tumor stage and poorer prognosis (Lab Investigation 2003, 83, 731-742). However, the authors show a better overall survival in patients with high NK-1R levels, while at the same time, NK1R was significantly decreased in PDAC tissue compared to normal tissue. These results are contradictory to what one might expect and with those previously published. This should be discussed.

- NK-1R antagonist aprepitant significantly reduces growth in PDAC cell lines and cancer stem cell-like cells.  The IC50 value for PDAC cells over a time period of 24 h. The lowest sensitivity to AP was detected in PSCs (32 µM), followed by Panc-1 (30 µM), HuP-T3 (29 µM), DanG (26 µM), and MIA PaCa-2 (19 µM). It must be indicated if (32 µM) it is CAPAN-1. Moreover, it has been reported previously the IC50 of aprepitant value for PDAC cell lines PA-TU- 8902 (31.2 µM) and CAPAN-1 (27.4 µM) respectively (Inves New Drugs 2010, 28, 187-193). The paper above mentioned should be included and discussed.

- Furthermore, it is known that aprepitant a designer drug is a highly selective non-peptide NK-1R antagonist that binds to the human NK-1R. In radio-ligand binding assays, aprepitant was approximately 3000-fold more selective for the human cloned NK-1R (IC50 = 0.1nM) versus the human cloned NK-3R (IC50 = 300nM) and 45,000-fold versus the human cloned NK-2R (IC50 = 4500 nM). Obviously, the antitumor action of aprepitant is derived from its activity on NK-1R which is overexpressed in cancer cells, including pancreatic cancer cells. However, you suggest a potential involvement of a PDAC-specific mechanism, where aprepitant binds with higher affinity to glucocorticoid receptors in cases of extreme downregulation of the NK-1R with concurrent upregulation of glucocorticoid receptors. Is there a published article on the binding of aprepitant to glucocorticoid receptors? Why do they suggest this idea? It is necessary to provide bibliographic references that support this hypothesis.

- Additionally, it has been reported that SP at 5 to 100 nM elicits pancreatic cancer cells proliferation (Lett Drug Des Discov 2006, 3, 323-329). SP is the natural ligand of NK-1R. In competition assay, NK-1R antagonist L-733,060-induced growth inhibition was partially reversed by the administration of a nanomolar dose of exogenous SP. The NK-1R antagonist L-733,060 inhibited pancreatic cells proliferation via interaction with its receptor. This means that pancreatic cancer cells must express NK-1R for that SP peptide or NK-1R antagonists to have effect. Indeed, theses pancreatic cancer cell lines have been demonstrated that express isoforms NK-1R (Lett. Drug Des. Discov. 2006, 3, 323-329).

Moreover, as well you say, targeting of the NK-1R with its antagonist aprepitant resulted in anti-tumor effects in pancreatic cancer. To supported this claim, it is necessary to demonstrate that pancreatic cancer cells express NK-1R as previously published (Lab Investigation 2003, 83, 731-742)

- Based in the above mentioned, it is necessary to perform a western blot on the five pancreatic cancer cell lines studied for to determine the expression of NK-1R and to repeat the real-time quantitative RT-PCR in the five pancreatic cells lines for to determine the expression of NK-1R.

Author Response

(The authors gave the same response as above.)

Round 2

Reviewer 2 Report

Now all is OK

Author Response

We thank the reviewer for this positive note as well as for his timely and fast work.